# What mechanisms mediate prior probability effects on rapid-choice decision-making?

**Rohan Puri** [1]*, **Mark R. Hinder**[1], **Andrew Heathcote**[2,3]

**1** Sensorimotor Neuroscience and Ageing Research Group, School of Psychological Sciences, College of Health and Medicine, University of Tasmania, Hobart, Australia, **2** School of Psychology, University of Newcastle, Newcastle, Australia, **3** Department of Psychology, University of Amsterdam, Amsterdam, The Netherlands

* rohan.puri@utas.edu.au

## Abstract

Rapid-choice decision-making is biased by *prior probability* of response alternatives. Conventionally, prior probability effects are assumed to *selectively* affect, response threshold, which determines the amount of evidence required to trigger a decision. However, there may also be effects on the rate at which evidence is accumulated and the time required for non-decision processes (e.g., response production). Healthy young ($n = 21$) and older ($n = 20$) adults completed a choice response-time task requiring left- or right-hand responses to imperative stimuli. Prior probability was manipulated using a warning stimulus that informed participants that a particular response was 70% likely (i.e., the imperative stimulus was either congruent or incongruent with the warning stimulus). In addition, prior probability was either fixed for blocks of trials (block-wise bias) or varied from trial-to-trial (trial-wise bias). Response time and accuracy data were analysed using the racing diffusion evidence-accumulation model to test the selective influence assumption. Response times for correct responses were slower on incongruent than congruent trials, and older adults' responses were slower, but more accurate, than young adults. Evidence-accumulation modelling favoured an effect of prior probability on both response thresholds *and* nondecision time. Overall, the current results cast doubt on the selective threshold influence assumption in the racing diffusion model.

## Introduction

Humans live in a dynamic, ever-changing, world requiring navigation, in a timely manner, through different possible decision outcomes to achieve goal-directed behaviour. Often, these possible decision outcomes–conceptualised here as the time-dependent accumulation of sensory evidence towards a response threshold–are influenced by factors such as different payoffs or prior probabilities [1]. For example, when getting ready to return a serve in tennis, a player may weigh a possible forehand return more than a backhand return based on the known higher prior probability of the opponent serving to their forehand. Then, if the opponent does indeed serve to the forehand, the player would have a quicker reaction-time (and thus higher chance of making a good return) than if the opponent unexpectedly served to their backhand.

**Data Availability Statement:** All data and analyses files are available from the Open Science Framework database (https://osf.io/yh2rp/).

**Funding:** RP was supported by an Australian Government Research Training Program (RTP)

Scholarship, MRH by Australian Research Council's Discovery Scheme via a Future Fellowship (FT150100406) and Discovery Project (DP200101696), and AH by Australian Research Council's Discovery Project (DP200100655). The funders had no role in study design, data collection and analysis, decision to publish, or preparation of the manuscript.

**Competing interests:** The authors have declared that no competing interests exist.

This effect of prior probability on decision-making processes is classically thought to *selectively* influence response thresholds, such that the more likely outcome has a lower response threshold, and the less likely outcome has a higher response threshold. In the previous example, the response threshold for a forehand return would be lower than that for a backhand return, thus facilitating a faster reaction-time. This canonical effect of prior probability on response thresholds has been observed in domains of perceptual decision-making [2–9] as well as recognition memory [8, 10, 11].

Recently, however, the selective influence of prior probability on response thresholds has come under question. Specifically, it has been claimed that prior probability can also bias sensory evidence [9, 12]. Given this potential non-selectivity of prior probability on decision processes, one may also wonder if, and how, prior probability affects *nondecision*-based processes such as stimulus encoding and response execution. Indeed, mediation by more than one mechanism occurs for another canonical finding in decision-making research, the 'speed-accuracy trade-off'. The trade-off is named as such because participants are often quicker, but less accurate, when responding under instructions or feedback/payoffs that emphasise speed, and are slower, but more accurate, when responding under instructions or feedback/payoffs that emphasise accuracy. The effect of the speed-accuracy trade-off on decision-based processes was thought to be selectively reflected in response *caution*, mediated by an increase in thresholds for all responses, such that participants are more cautious when responding with an accuracy, rather than speed, emphasis [2, 13, 14]. However, in recent times, it has been demonstrated that the effect of the speed-accuracy trade-off is non-selective. Specifically, this non-selectivity of decision-based processes is observed in the quality of sensory evidence [15–22] and, relevant to the current study, in nondecision-based processes as well, with speed emphasis reducing the time for these processes to be executed [23–25].

Accordingly, given the potential non-selectivity of prior probability on decision-based processes and the effect of other classical manipulations on nondecision-based processes, the primary aim of the current study was to investigate the effect of prior probability on nondecision-based processes. To assess the generality of any such effects, our study incorporated two further manipulations. The first pertains to the *flexibility* with which prior probability information is utilised. Specifically, the more probable response was either kept constant across a block of trials ('block-wise bias'; [2, 3, 5, 7, 8, 10, 11]) or varied from trial-to-trial ('trial-wise bias'; [4, 9]). Recently, a direct comparison between these two biasing procedures revealed that effects on response thresholds were greater for block-wise compared to trial-wise biasing [26]. Given this effect on decision-based processes, the current study aimed to elucidate whether any effect of prior probability on nondecision-based processes also differed between block- and trial-wise biasing procedures. The second is a manipulation that addresses the influence of healthy ageing on the utilisation of prior probability information. Garton and colleagues [26] demonstrated that older adults differ in their use of prior probability information and change their thresholds to a greater degree than young adults, at least with a block-wise bias manipulation. Given this effect on decision-based processes, the age-related decline in motor function [27], and the known difference in nondecision-based processing between young and older adults [28], the current study aimed to elucidate whether any effect of prior probability on nondecision-based processes varied due to healthy ageing as well as its interaction with the two, aforementioned, biasing procedures.

To this end, in the current study of binary choice, the evidence-accumulation process for speeded decisions was characterised using a "sequential sampling model", namely the *racing diffusion model* [29, 30]. Specifically, as illustrated in Fig 1, each accumulator–representing a decision associated with a 'left' or 'right' response–begins with an initial amount of evidence and accumulates further evidence in a noisy manner. This process of evidence-accumulation

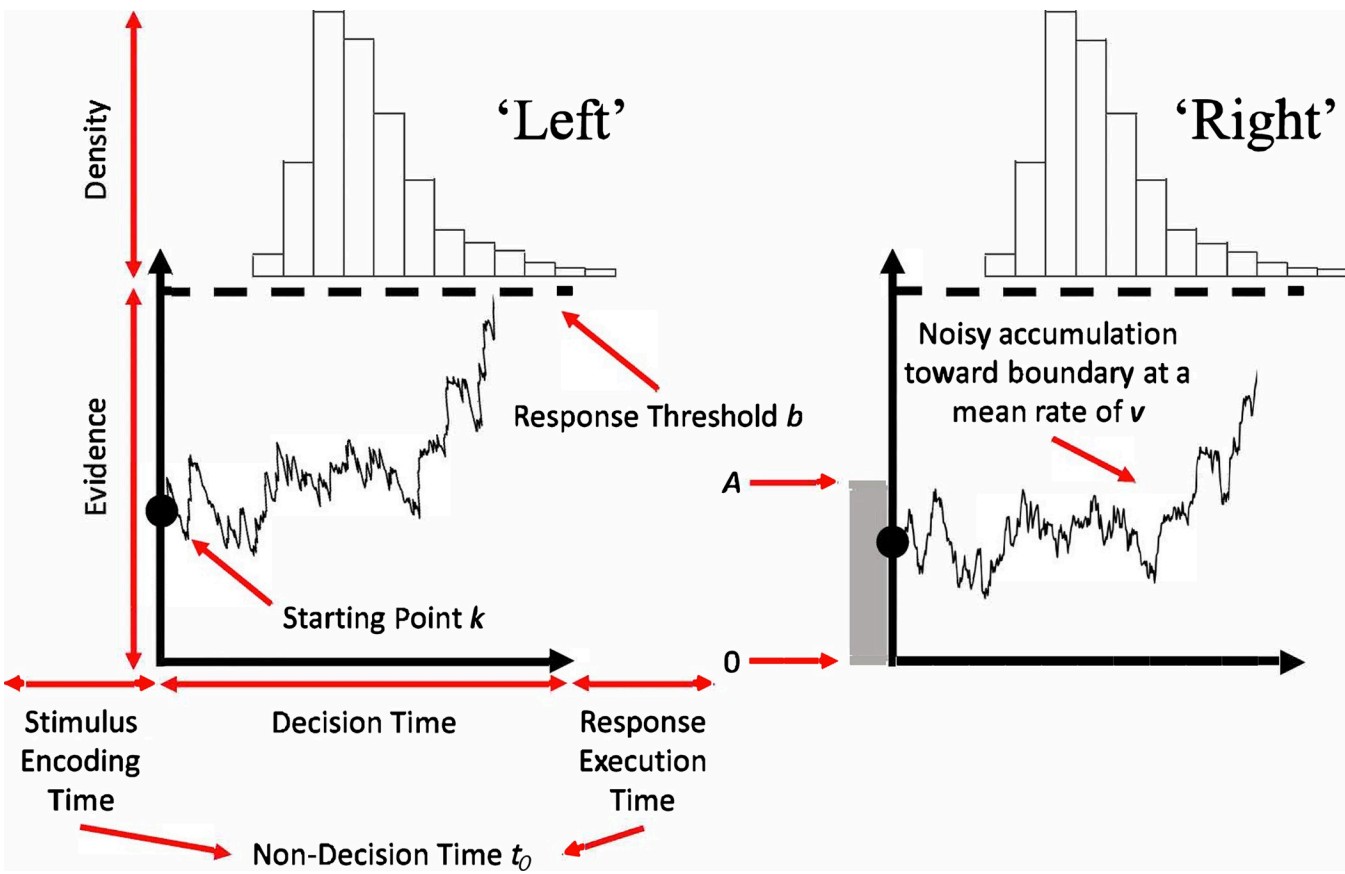

**Fig 1. Racing diffusion model.** Two-choice model representing a decision between left and right responses. Each accumulator begins with an initial amount of evidence ($k$), drawn from a uniform distribution $[0, A]$, that increases at a noisy rate, with mean $v$, towards the response threshold ($b$). Decision time–the time taken for the winning accumulator (in this example, the Left response) to reach its respective threshold, along with nondecision time ($t_0$)–time taken for stimulus encoding and response execution components–determine the overall response time. Adapted with permission (not part of governing open access license) from Springer Nature Customer Service Centre GmbH: Springer Nature, Psychonomic Bulletin & Review, Tillman, G., Zandt, T. Van, & Logan, G. D. Sequential sampling models without random between-trial variability: the racing diffusion model of speeded decision making, *The Psychonomic Soceity, Inc. (2020).*

stops when the first accumulator reaches its threshold, with RT corresponding to the time taken to do so (decision time) in addition to stimulus encoding and response execution time (i.e., nondecision time). In the current study, our use of this racing diffusion model makes two key assumptions. Firstly, it was reasoned that (any) effects of prior probability on the nondecision time parameter was reflective of effects on the response execution time, and not the stimulus encoding time, component. Given the identical optical properties of the stimuli in the current study (see 'Experimental procedure' section) and that stimulus encoding extracts decision-relevant evidence, it is highly unlikely that the stimulus encoding time component is different for the left vs. right accumulators. In contrast, differential priming of left vs. right responses based on prior probability is highly plausible, resulting in differences in the response execution time component. Indeed, based on transcranial magnetic stimulation studies using electromyography, it has been demonstrated that the corticospinal tract projecting to the effector corresponding to the higher probability response exhibits greater excitation and reduced inhibition than the effector corresponding to the lower probability response [31, 32].

However, one may argue that the different priming of left vs. right responses, and more broadly decision-based processes as well, may be influenced by other mechanisms besides

prior probability. Specifically, the provision of a warning signal ('prime'), regardless of its validity, prior to the imperative signal ('target') might invoke *response priming* with the visual similarity between the warning and imperative signal possibly invoking *perceptual priming* (see Fig 2 and 'Experimental procedure' section). Response priming in the current study was tested with the presence of 'catch' trials (i.e., trials where the warning signal was *not* followed by an imperative signal) such that a negligible influence of response priming would be reflected by participants correctly withholding their responses on catch trials (i.e., lower false alarm rate). Perceptual priming was tested by ensuring that the warning signal prime and the imperative signal target utilized the same symbol (arrows as per Fig 2), with any perceptual priming effects biasing sensory evidence and being reflected as superior model fits of consistent evidence-accumulation parameters ($v$ parameter in Fig 1).

Secondly, our model assumes that observed responses are based on a stimulus-driven decision-making process rather than a "fast guess" process in which stimulus onset triggers a guessed response that is chosen prior to stimulus onset [3, 33, 34]. Fast guesses are known to occur on a subset of trials in bias manipulations of prior probability and payoffs, with the fast guess response usually favouring the more probable and higher payoff response [3, 33]. However, given that this is especially the case in *extreme* levels of bias where it is optimal to allow stimulus onset to trigger the fast guess response [3, 35], the current study utilised a moderate level of bias to minimize fast guesses. Specifically, as illustrated in Fig 2, the warning signal indicated, prior to stimulus onset, a 70% bias towards a particular response. A second purpose of including catch trials was such that if participants utilised a fast-guessing strategy, one would expect a higher false alarm rate (and vice versa). Furthermore, an exponentially-weighted moving average RT filter was utilised to determine, for each participant separately, the cut-off point below which RTs were reflective of a fast-guessing strategy (i.e., 50% accuracy in responses). Based on these procedures and the evidence (see 'Data processing' section), we are confident in our assumption that filtered responses are reflective of a stimulus-driven decision-making process.

In summary, using a moderately biased choice RT task and the racing diffusion model, the current study investigated the effect of prior probability, in the different biasing procedures, on response execution across the lifespan. We investigated whether: 1) response thresholds are lower and/or 2) mean accumulation rate higher and/or 3) response execution time shorter for the accumulator corresponding to the higher probability stimulus. We further investigated whether these effects were more prominent under the block-wise than trial-wise biasing procedure, and in older compared to young adults.

## Materials and methods

### Participants

Twenty-one healthy young adults (mean age = 26.76 years; SD = 5.85 years; range = 19 to 40 years; all but two self-declaring right-handed dominance) and 20 healthy older adults (mean age = 67.95 years; SD = 5.12 years; range = 61 to 84 years; all but one self-declaring right-handed dominance) were recruited between 2017 and 2018 from the local university and the community. The Mini-Mental State Examination [36] was used to screen older adults for cognitive integrity and everyone scored within the normal range (score ≥ 26; [37]). Prior to participating in the study, all participants provided written informed consent. De-identified data were used for all analyses with re-identification only possible through a secure master coding file so that individual data could be withdrawn prior to publication if requested by a participant. The study was approved by the Tasmanian Human Research Ethics Committee Network and conducted in accordance with the Declaration of Helsinki.

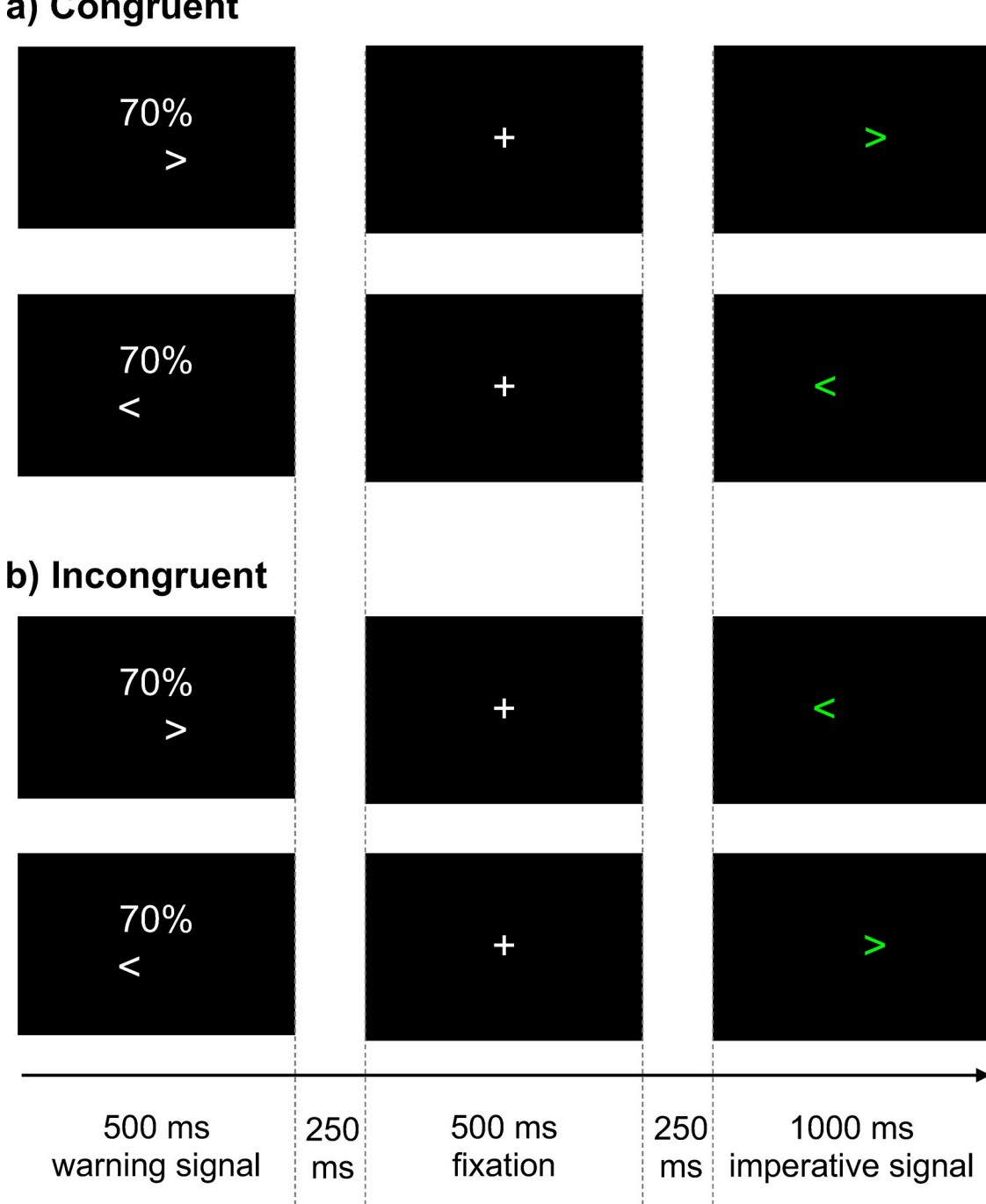

**Fig 2. Experimental trials.** On a computer monitor, participants were presented a 500 ms warning signal which indicated a 70% probability of the upcoming imperative signal pointing either to the right (>) or left (<). Following the warning signal, a blank screen (250 ms), a fixation cross (500 ms), and another blank screen (250 ms) were presented. Following these, a 1000 ms green coloured imperative signal (> or <) was presented either in the same (**a**: congruent trials) or opposite (**b**: incongruent trials) direction to the warning signal. Participants were required to respond with a button consistent with the direction of the arrow in the imperative signal.

## Experimental procedure

On separate days, at least 24 h apart, participants attended two sessions (1.5–2.5 h duration). One session only involved the behavioural task, and the other session involved the behavioural task as well as non-invasive brain stimulation which has been published previously [32]. Given the known influence of brain stimulation on RT [38–40], data from the session only involving the behavioural task is analysed here.

Participants undertook a choice RT task in which button press responses to simple visual stimuli, using their left or right index fingers, were required. The likelihood of a particular response was conveyed using a warning stimulus presented prior to the imperative stimulus. Thus, on each trial, a warning signal, "70% >" or "70% <", was first presented for 500 ms indicating the true probability of the upcoming imperative signal pointing in the same direction. Following this, a blank screen (250 ms), a central fixation cross (500 ms), and another blank screen (250 ms) were presented in the listed temporal order. Following these, a 1000 ms green imperative signal, pointing either to the right, ">", or to the left, "<", was presented on a computer monitor. Participants were instructed to respond, as quickly and accurately as possible, using their right (to ">") or left (to "<") index finger by pressing buttons on a USB response pad (The Black Box ToolKit, Sheffield, UK). Thus, trials were classified as either *congruent* trials (70% of trials)–where the warning and imperative signals pointed in the same direction–or *incongruent* trials (30% of trials)–where the warning and imperative signals pointed in opposite directions, as depicted in Fig 2. Participants were provided with feedback 750 ms after every trial, composing of either a) the RT (in seconds) if a correct response was made, or b) "Incorrect press" if an incorrect response was made, or c) "Missed" if no response was made. Consequently, the total trial duration was 3250 ms and the inter-trial interval was set to 750 ms. PsychoPy2 v1.84.2 was used to present all experimental stimuli [41].

Twenty blocks of 65 trials each were presented using *block-wise* and *trial-wise* bias procedures to investigate differences in bias control. Specifically, for block-wise biasing, the warning signal remained the same across the whole block of trials and hence the more likely response option across the whole block of trials remained the same. For trial-wise biasing, the warning signal changed pseudorandomly from trial-to-trial in a block and consequently the more likely response option changed pseudorandomly from trial-to-trial in a block, with an equal number of left- and right-handed response options in a block of trials. Overall, participants completed 10 blocks each of the block-wise and trial-wise bias, counterbalanced across participants. Five trials in each block were 'catch' trials (i.e., no imperative signal followed the warning signal) where participants were required to withhold their response. These catch trials were implemented to ensure that participants were not anticipating the direction and timing of the imperative signal [32, 42]. Of the remaining 60 trials, 42 were congruent trials and 18 were incongruent trials (considering that the warning signal was always accurate and always indicated a 70% probability).

## Data processing

For one participant two blocks of data could not be collected due to technical difficulties. Furthermore, prior to model fitting, catch trials (on average, the false alarm rate was low, with participants correctly withholding their response on 94.9% of catch trials suggesting negligible response priming effects and that participants did not utilize a wholly anticipatory 'fast guessing' strategy) and trials on which no response was made (average of 1.5% across all participants) were excluded.

In addition to utilising catch trials and ensuring that the prior probability bias manipulation was not too extreme, the inclusion of any fast guess trials in the final sample was further

minimized by applying two types of RT filters. Firstly, RTs less than 200 ms were discarded. Secondly, for each participant and each bias type, RTs less than 'cut-off' RTs–found using an exponentially-weighted moving average (EWMA) of accuracy after sorting by RT (starting point = 0.5; weighting alpha parameter = 0.1; cut-off accuracy threshold = 0.6)–were discarded [43, 44]. Overall, 1265 of 48,306 trials were removed (2.6%) based on these two procedures. Of the 1265 trials, 819 trials were classified as having RTs below 200 ms, 151 trials were classified as having RTs below the cut-off determined by the EWMA procedure, and 295 trials were classified as meeting both the aforementioned criteria. Consequently, due to these various procedures, histograms of RTs where the imperative signal was congruent with the biasing warning signal (i.e., the condition where fast guesses would be most apparent) showed no evidence of participants utilising a fast-guessing strategy (see S3 Fig).

### Racing diffusion model

**Model fitting.** The racing diffusion model utilised in the current study, as illustrated in Fig 1, includes several parameters for each of the two accumulators (one for a left and one for a right response). Mean evidence-accumulation rate, $v$, was parameterised separately for the (correct) accumulator that matched the stimulus ('matching $v$') and the (incorrect) accumulator that mismatched the stimulus ('mismatching $v$'). In addition, to incorporate bias effects, $v$ was parameterised as a 'bias-right' accumulation rate parameter, $v_{br}$, by scaling to the minimum accumulation rate and adding it to the right accumulator (and subtracting from the left accumulator). Thus, $v_{br}$ would range between +1, indicating maximum right bias, and -1, indicating maximum left bias, with 0 indicating no bias. For example, if the rates for left and right accumulator are $v_{left}$ and $v_{right}$ before the addition of bias, and denoting the minimum of these two values as $d_v$ (typically the rate for the mismatching accumulator), the rates after the addition of bias (with subscripts "right,bias" and "left,bias") become:

$$v_{right,bias} = v_{right} + d_v * v_{br} \tag{a}$$

$$v_{left,bias} = v_{left} - d_v * v_{br} \tag{b}$$

Note that scaling $v_{br}$ by $d_v$ ensures that the rates remain non-negative.

Threshold and nondecision time were parameterised as $b$ and $t_0$, respectively, for every bias direction (left and right) and accumulator (left and right) combination. Given that accumulators raced with a positive evidence-accumulation rate towards a positive threshold, RTs are described by a shifted-Wald distribution, with the shift equal to the nondecision time component [29, 30]. Of note, i) the start-point noise parameter, $A$, was kept constant across all conditions, and ii) threshold results are reported in terms of $B = b - A \geq 0$, enforcing the constraint $b \geq A$ (i.e., accumulation cannot begin above the threshold) and since $A$ was constant across all conditions, differences in $B$ reflect pure threshold effects.

**Sampling.** Hierarchical Bayesian techniques, using an MCMC algorithm as implemented in the *Dynamic Models of Choice* R functions [45], were utilized to obtain the entire posterior probability distribution of the parameters. First, for each participant separately, posterior distributions were estimated using multiple chains (precisely, three times the number of parameters). Diffuse and independent normal priors were set for all parameters. Specifically, means and standard deviations for $B$, *matching $v$*, *mismatching $v$*, $v_{br}$, and $t_0$ were set to 2, 3, 1, 0, 0, and 2, 4, 4, 1, and 0.5, respectively. A lower bound of 0 was used for all parameters except for $v_{br}$ (set to -1) with no upper bound except for $v_{br}$ and $t_0$ having an upper bound of +1 and 1000 ms, respectively.

Individual fits starting from points sampled from the priors provided start-points for hierarchical models, which assumed independent normal population distributions, and were fit separately for each age group (young and older) and bias procedure (block-wise and trial-wise). Hyper-priors for population mean parameters were assumed to be the same independent normal priors as for individual fits. Hyper-priors for population standard deviations were exponential with a mean of 1 for all parameters. Each MCMC chain during hierarchical fitting was thinned by 10 (i.e., only one iteration in every 10 was retained) and a 5% probability of migration used until all chains had similar posterior likelihoods. Hierarchical sampling continued until Gelman's multivariate potential scale reduction factor was less than 1.1 [46], indicating mixing, stationarity, and convergence. Lastly, a final set of 250 iterations per chain (after thinning) was run, with the resulting samples used in further analyses. Convergence was confirmed by visual inspection of trace plots of the final samples.

**Model selection.**  For each age group and bias type combination, candidate hierarchical models were constructed where bias (i.e., warning signal biasing a left- or right-hand response) affected one or more of the threshold, evidence-accumulation rate, and nondecision time parameters, resulting in seven different models. Without a bias effect there are two threshold parameters (one for the left accumulator and one for the right accumulator), two rate parameters (one for the accumulator matching the stimulus and one for the mismatching accumulator) and two nondecision time parameters (one for the left accumulator and one for the right). In each case this number doubles with an added bias effect with different values estimated depending on whether the bias is to the left or right. We constructed acronyms for each model from the bias-affected parameters. In three models, each with 8 parameters, the bias affects: i) only threshold ($B$ model), ii) only accumulation rate ($v$ model), iii) only nondecision time ($t_0$ model). In three more models, each with 10 parameters, bias affects: iv) threshold and accumulation rate ($Bv$ model), v) threshold and nondecision time ($Bt_0$ model), vi) accumulation rate and nondecision time ($vt_0$ model). In the seventh model, with 12 parameters, bias affects vii) threshold, accumulation rate, and nondecision time ($Bvt_0$ model). The 7 hierarchical models were fit for each age group and bias type combination, resulting in a grand total of 28 hierarchical models. Model selection was conducted using the *deviance information criterion* (DIC), which penalizes goodness of fit measured by likelihood with a complexity penalty [47], where smaller values indicate a preferred model. As with most such information criteria, a difference of 10 units or more indicates a substantial preference [45].

Converging evidence on the relative importance of each parameter type in explaining the effect of bias was evaluated in several ways. First, we compared DIC values over the models with only one bias effect, with the best (lowest DIC) model indicating its parameter type is most influential. Second, we compared models with two bias effects, with the parameter type being omitted in the best model indicated as least influential. Finally, for the overall best (i.e., the model with the lowest DIC value among all seven) and second-best models, the relative importance of each parameter type in simulating bias effects on two key behavioural measures of bias, response probability and correct RT, were evaluated by removing the bias effect on each parameter type while keeping it on the remaining parameter types [48]. For rate bias $v_{br}$ was set to 0 for each posterior parameter sample. For $B$ and $t_0$ the four parameter estimates were reduced to two for each posterior sample by averaging over the bias manipulation. Random samples from the transformed posteriors were then used to generate 100 posterior predictive data sets, each consisting of a RT and choice for each trial, condition, and participant. The resulting distribution over 100 data sets of differences between incongruent and congruent conditions captures the model's average prediction and uncertanty about the effects averaged over trials, conditions, and participants of parametrer types that retained bias effects. We report the median and 95% credible interval of the bias effect predicted by the reduced model

divided by the effect predicted by the full model. When this ratio is close to 1, the removed parameter effect has little impact on the bias effect, whereas ratios less than 1 indicate that the removed parameter has a substantial role in explaining the predicted bias. Occasionaly the ratio can be greater than 1, indicating a trade-off where the removed parameter masks a larger bias caused by the remaining parameter effect(s).

## Results

The data and analyses for the current study are available at https://osf.io/yh2rp/.

### Model selection

In the trial-wise bias type, for both young and older adults, the model where bias varied over $B$, $v$, and $t_0$ ($Bvt_0$ model) had the smallest DIC whereas in the block-wise bias type the model where bias varied over $B$ and $t_0$ ($Bt_0$ model) had the smallest DIC (Table 1). These two models were clearly better than the others in all cases. For scenarios where bias was allowed to vary over two parameters, the $Bt_0$ model had the smallest DIC for all age group and bias type combinations, indicating that accumulation rate had the least role in explaining bias effects. For scenarios where bias was only allowed to vary over a single parameter, the threshold only, $B$, model had the smallest DIC for all combinations, indicating it is the most influential parameter type.

Although the mean $B$ and $t_0$ parameter values for the two best overall models were in the predicted directions (i.e., lower $B$ and $t_0$ values for congruent compared to the incongruent parameters; S1 Table), there was a lack of consistency for $Bvt_0$ model's accumulation rate estimates, with 5 out of 8 values causing a *negative* bias effect (i.e., better performance in the incongruent condition, see S2 Table), suggesting overfitting by the $Bvt_0$ model and inconsistent perceptual priming effects.

This interpretation is supported by posterior predictive analysis of the $Bvt_0$ model. As shown in Table 2, the predicted correct RT bias effect (i.e., incongruent correct RT–congruent correct RT) and the predicted accuracy bias effect (i.e., incongruent proportion correct–congruent proportion correct) were very close to the observed effects. With no threshold bias effect there was substantial misfit relative to the full model, with anywhere from half to all of the biasing in accuracy and correct RT removed. Omitting the nondecision time bias effect had a more moderate effect causing a reduction of about one third in correct RT bias (nondecision time has no effect on accuracy). In contrast, omitting the accumulation rate bias effect led to inconsistent results, sometimes causing either relatively small increases or decreases in the predicted bias effects. These results suggest that bias effects were primarily driven by

**Table 1. Model fit.**

|  | Young adults Block-wise bias | Young adults Trial-wise bias | Older adults Block-wise bias | Older adults Trial-wise bias |
|---|---|---|---|---|
| $v$ | 708 | 601 | 823 | 694 |
| $t_0$ | 299 | 211 | 228 | 163 |
| $B$ | 217 | 121 | 135 | 140 |
| $vt_0$ | 114 | 92 | 207 | 99 |
| $Bv$ | 182 | 122 | 123 | 44 |
| $Bt_0$ | 0 (-38798) | 5 | 0 (-28270) | 38 |
| $Bvt_0$ | 6 | 0 (-38708) | 9 | 0 (-28525) |

Model fit measures (values represent differences in DIC relative to the model with the smallest DIC, indicated with a 0 and the DIC value in parentheses) for models where bias was allowed to vary over unique combinations of threshold (B), accumulation rate (v), and nondecision time (t₀) for all age group (young and older adults) and bias type (block-wise and trial-wise bias) combinations, separately.

**Table 2. Posterior predictive analysis for $Bvt_0$ model.**

| | | | Omitted effect | | | | |
| | | | $v$ | | $B$ | | $t_0$ |
| | RT | ACC | RT | ACC | RT | ACC | RT |
|---|---|---|---|---|---|---|---|
| **YB** | -3 [−5, −1] | 0.01 [0.001, 0.03] | 0.87 [0.75, 1.01] | 0.76 [0.49, 1.03] | 0.47 [0.30, 0.65] | 0.47 [0.20, 0.83] | 0.66 [0.54, 0.79] |
| **YT** | -2 [−5, 1] | 0.01 [0.004, 0.02] | 0.99 [0.84, 1.14] | 0.95 [0.62, 1.42] | 0.34 [0.12, 0.59] | 0.21 [-0.07, 0.58] | 0.67 [0.51, 0.85] |
| **OB** | 1 [−2, 5] | 0.01 [0.002, 0.02] | 1.00 [0.90, 1.13] | 0.97 [0.76, 1.31] | 0.31 [0.16, 0.42] | 0.13 [-0.06, 0.28] | 0.69 [0.54, 0.86] |
| **OT** | 1 [−3, 5] | 0.01 [-0.001, 0.02] | 1.17 [1.04, 1.31] | 1.41 [1.07, 2.05] | 0.19 [-0.02, 0.38] | -0.03 [-0.27, 0.25] | 0.61 [0.46, 0.77] |

For each age group (Y: young adults; O: older adults) and bias type (B: block-wise bias; T: trial-wise bias) combination, differences between the $Bvt_0$ model predicted bias effect and observed bias effect (response time in milliseconds as RT and proportion correct for accuracy as ACC) as well as the ratios of the bias effect with threshold, B, accumulation rate, v, or nondecision time, $t_0$, omitted are reported for both, correct response-time (RT) and accuracy (ACC). Note, numbers within square brackets represent the lower and upper bounds of the 95% credible intervals and for $t_0$ only RT is presented.

threshold and secondarily by nondecision time. Posterior predictive analysis of the $Bt_0$ model (Table 3) supports this conclusion. The quality of predictions for both accuracy and correct RT bias effects is comparable to the $Bvt_0$ model.

Considering all the above converging evidence, the $Bt_0$ model provides the best account of all conditions. Observed distributions of RT, for both correct and error responses, and accuracy, along with predictions for the $Bt_0$ model, are illustrated in Figs 3 and 4, respectively. Overall, as expected, older adults had slower but more accurate responses than young adults. RTs were slower on correct incongruent than correct congruent responses, whereas the opposite pattern was observed for error responses. The model captures these trends and provides a good quantitative account, with the observed data falling within the model-predicted 95% credible intervals. However, some over-estimation occurred for error RTs along with a small (~1%) under-estimation of the proportion of correct incongruent responses. This misfit was not addressed by the $Bvt_0$ model (see S1 and S2 Figs).

## Parameter tests

Parameter tests were conducted on the $Bt_0$ model for thresholds, $B$, and nondecision time, $t_0$, parameters separately. Table 4 presents the median estimates for $B$ and $t_0$ at every combination

**Table 3. Posterior predictive analysis for $Bt_0$ model.**

| | | | Omitted effect | | |
| | | | $B$ | | $t_0$ |
| | RT | ACC | RT | ACC | RT |
|---|---|---|---|---|---|
| **YB** | -3 [−5, −1] | 0.01 [0.00, 0.02] | 0.22 [0.09, 0.34] | 0.14 [-0.01, 0.31] | 0.74 [0.62, 0.89] |
| **YT** | -2 [−5, 0] | 0.01 [0.00, 0.02] | 0.32 [0.18, 0.49] | 0.13 [-0.07, 0.31] | 0.66 [0.51, 0.85] |
| **OB** | 2 [−3, 6] | 0.01 [0.00, 0.02] | 0.31 [0.19, 0.43] | 0.14 [-0.02, 0.34] | 0.68 [0.54, 0.85] |
| **OT** | 2 [−1, 6] | 0.01 [0.00, 0.02] | 0.56 [0.42, 0.71] | 0.46 [0.23, 0.70] | 0.43 [0.29, 0.61] |

For each age group (Y: young adults; O: older adults) and bias type (B: block-wise bias; T: trial-wise bias) combination, differences between the $Bt_0$ model predicted bias effect and observed bias effect (response time in milliseconds as RT and proportion correct for accuracy as ACC) as well as the ratios of the bias effect with threshold, B, or nondecision time, $t_0$, omitted are reported for both, correct response-time (RT) and accuracy (ACC). Note, numbers within square brackets represent the lower and upper bounds of the 95% credible intervals and for $t_0$ only RT is presented.

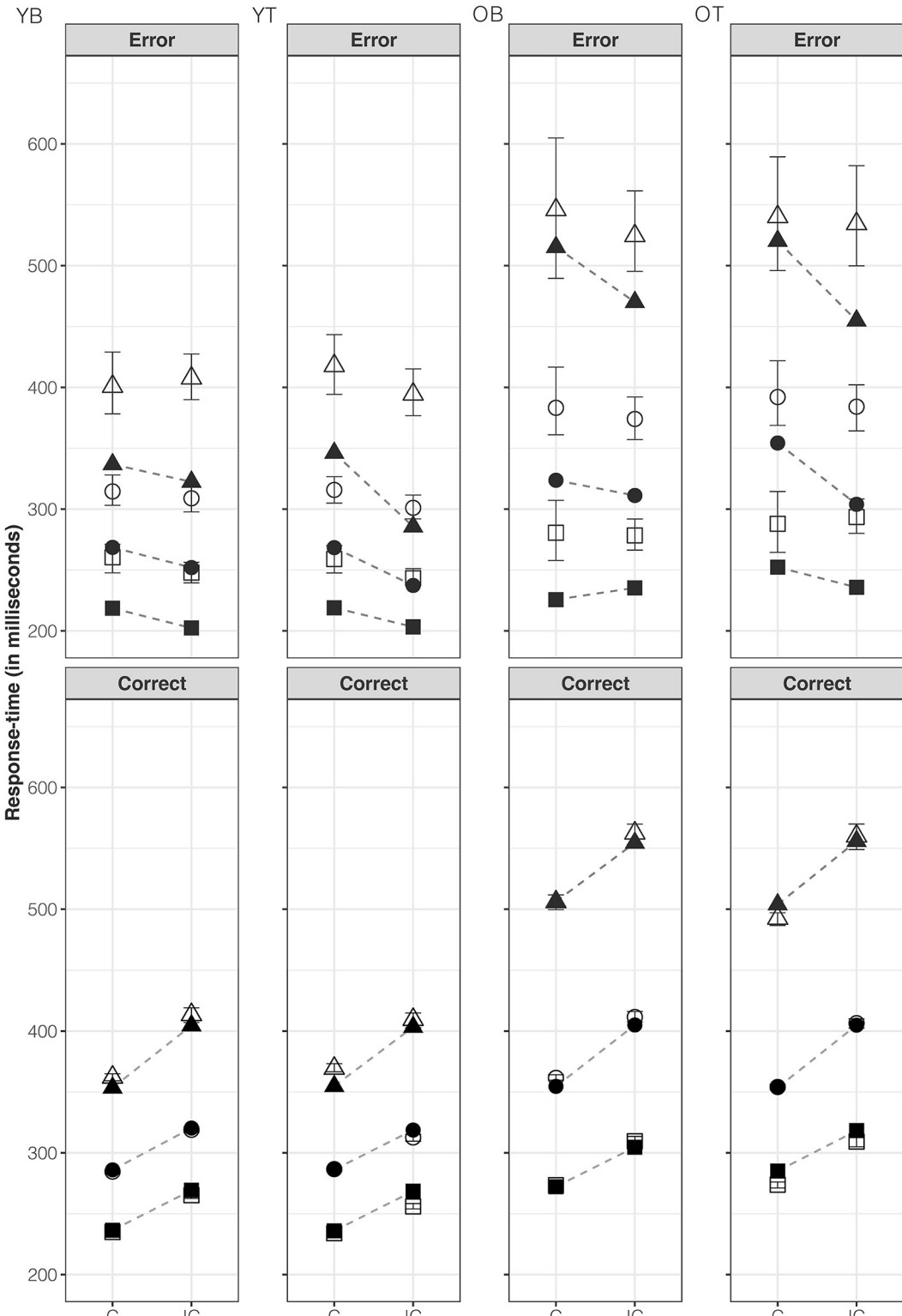

**Fig 3. Response time.** Observed (filled) and $Bt_0$ model-predicted (unfilled) 10th (square), 50th (circle), and 90th (triangle) error (top panel) and correct (bottom panel) response time percentiles plotted, in milliseconds, separately for each age group–young (Y) and older (O) adults–as well as each bias type–block-wise (B) and trial-wise (T). Observed data points between congruent (C) and incongruent (IC) trial types are connected via dashed lines. Error bars for model-predicted response times represent 95% credible intervals.

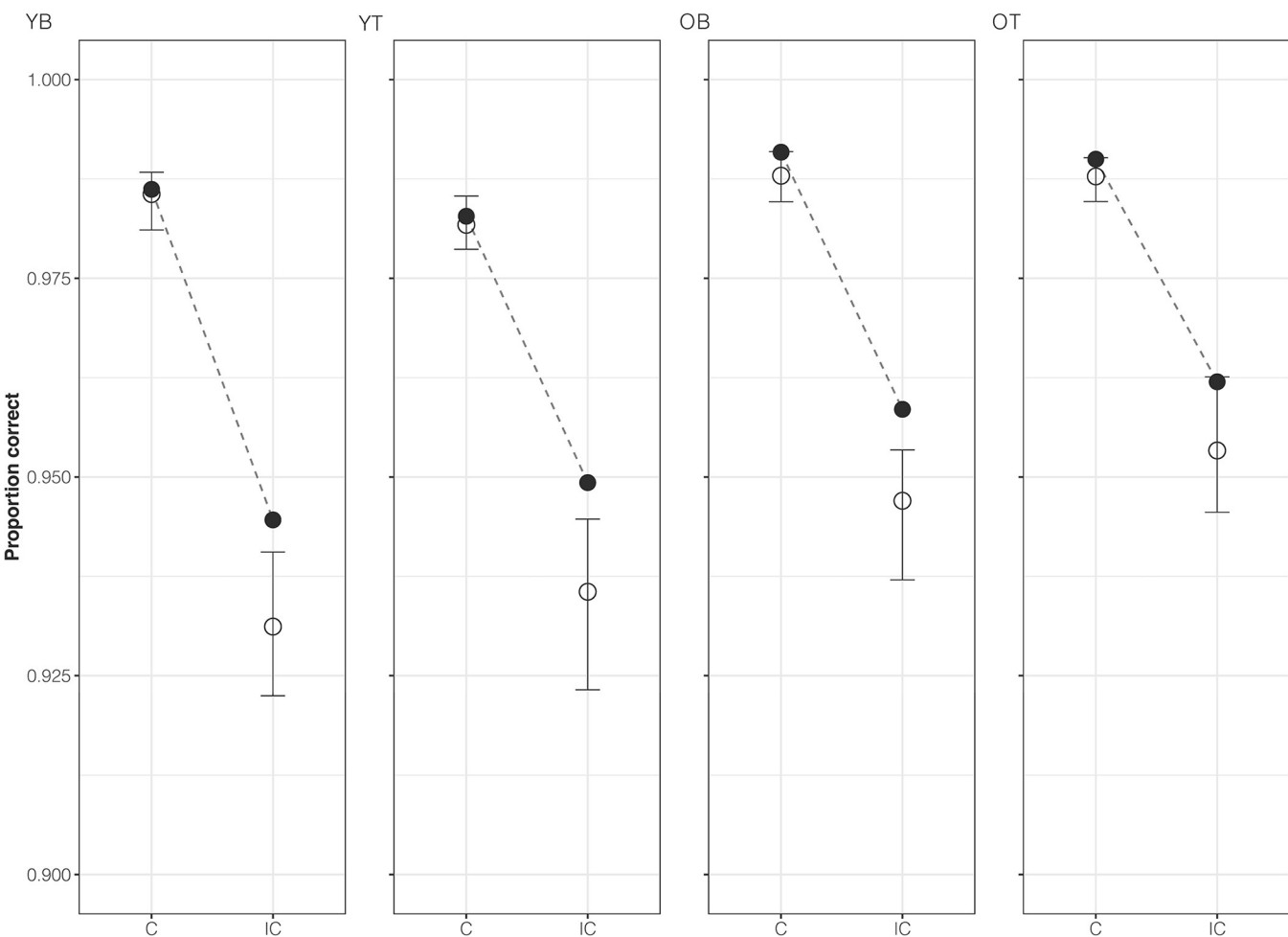

**Fig 4. Accuracy.** Observed (filled) and $Bt_0$ model-predicted (unfilled) proportion of correct responses plotted separately for each age group–young (A) and older (B) adults–as well as each bias type–block-wise (1) and trial-wise (2). Observed data points between congruent (C) and incongruent (IC) trial types are connected via dashed lines. Error bars for model-predicted proportion of correct responses represent 95% credible intervals.

of all experimental factors with Table 5 presenting the *differences* in median estimates (differences considered *credibly significant* if the 95% CI did not contain 0).

For *B*, in both young and older adults, credibly significant bias effects were observed in block- and trial-wise bias types with incongruent thresholds being greater than congruent thresholds. However, only for young adults was the threshold bias effect credibly greater in the block- than trial-wise bias type. Focussing on response *caution* (i.e., average of incongruent and congruent *B* parameters), comparing young and older adults, though caution was greater for older adults in both bias types, it was only credibly significant in the trial-wise bias type with this difference being credibly greater than in the block-wise bias type.

For $t_0$, in both young and older adults, credibly significant bias effects were observed in block- and trial-wise bias types with incongruent nondecision time being greater than congruent nondecision time. However, only for older adults was the nondecision time bias effect credibly greater in the trial- than block-wise bias type (14 ms [6–23]). Furthermore, comparing young and older adults, though the nondecision time bias effect was greater for older adults in both bias types, it was only credibly significant in the trial-wise bias type (20 ms [13–27]) with this difference being credibly greater than in the block-wise bias type (14 ms [3–24]).

**Table 4. Median parameter estimates.**

| | Young adults | | | | Older adults | | | |
|---|---|---|---|---|---|---|---|---|
| | Block-wise | | Trial-wise | | Block-wise | | Trial-wise | |
| | IC | C | IC | C | IC | C | IC | C |
| $B$ | 1.18 [1.14, 1.23] | 0.95 [0.93, 0.98] | 1.04 [1.01, 1.08] | 0.88 [0.86, 0.91] | 1.20 [1.16, 1.24] | 1.01 [0.98, 1.04] | 1.23 [1.19, 1.27] | 1.07 [1.04, 1.11] |
| $t_0$ | 176 [172, 180] | 167 [165, 169] | 184 [181, 188] | 175 [173, 177] | 216 [211, 221] | 201 [198, 204] | 218 [212, 223] | 188 [185, 192] |

Median estimates for threshold (B) and nondecision time ($t_0$, in milliseconds) parameters at every combination of age group (Young and Older adults), bias type (Trial-wise and block-wise), and trial type (Congruent and Incongruent) experimental factors. Values within square brackets represent the lower and upper bounds of the 95% credible intervals.

Focussing on the average speed of nondecision processing (i.e., average of incongruent and congruent $t_0$ parameters), comparing young and older adults, average nondecision processing speed was credibly greater for older adults in both bias types with this difference being credibly greater in the block- than trial-wise bias type.

In summary, older adults, compared to young adults, exhibited a greater nondecision time bias effect in trial- than block-wise bias type. In addition, older adults, compared to young adults, exhibited greater response caution in the trial- than block-wise bias type but exhibited longer average nondecision processing in the block- than trial-wise bias type.

## Discussion

The current study provides novel insights, from the racing diffusion model perspective, into the selective nature of prior probability effects. Specifically, besides the canonical effect of prior probability on response thresholds, a clear influence was also observed on a nondecision-based process of response execution, but not on accumulation rates. These observations were apparent across the lifespan (i.e., in both healthy young and older adults) as well as across both (i.e., block- and trial-wise) biasing procedures.

**Table 5. Differences in median parameter estimates.**

| | | Bias effect | | Response caution | Average speed of nondecision processes (ms) |
|---|---|---|---|---|---|
| | | $B$ | $t_0$ (ms) | | |
| **Young** | *Block-wise* | **0.23 [0.19, 0.27]** | **9 [4, 13]** | - | - |
| | *Trial-wise* | **0.16 [0.13, 0.19]** | **9 [5, 13]** | | |
| | *Block-wise vs. trial-wise* | **0.07 [0.02, 0.12]** | -0.2 [–6, 5] | **0.11 [0.07, 0.15]** | **-8 [–12, –5]** |
| **Older** | *Block-wise* | **0.19 [0.15, 0.23]** | **15 [9, 21]** | - | - |
| | *Trial-wise* | **0.15 [0.11, 0.20]** | **29 [23, 35]** | | |
| | *Block-wise vs. trial-wise* | 0.04 [-0.02, 0.10] | **-14 [–23, –6]** | **-0.044 [-0.087, -0.003]** | **6 [1, 10]** |
| **Young vs. Older** | *Block-wise* | 0.04 [-0.02, 0.09] | -7 [–14, 1] | -0.039 [-0.080, 0.003] | **-37 [–41, –33]** |
| | *Trial-wise* | 0.01 [-0.05, 0.06] | **-20 [–27, –13]** | **-0.19 [-0.23, -0.15]** | **-23 [–27, –19]** |
| | *Block-wise vs. trial-wise* | 0.03 [-0.05, 0.11] | **14 [3, 24]** | **0.15 [0.09, 0.21]** | **-14 [–20, –8]** |

Differences in median estimates to infer a) bias effects (incongruent–congruent) for threshold (B) and nondecision time ($t_0$, in milliseconds) parameters, b) response caution (averaged incongruent and congruent threshold parameters), and c) average speed of nondecision processes (averaged incongruent and congruent nondecision time parameters). Differences in block-wise, trial-wise, and between block-wise and trial-wise bias types are reported for young, older, and between young and older adults with certain differences, for b) and c), not theoretically possible ('-'). For block- and trial-wise bias types, positive values indicate a larger incongruent, than congruent, value, whereas for block-wise vs. trial-wise bias type and young vs. older adult comparisons, positive values indicate a greater bias effect of the former. Note, values within square brackets represent the lower and upper bounds of the 95% credible intervals and bolded values indicate differences being credibly different from 0.

In line with previous research in perceptual decision-making [2–9, 26] and recognition memory [8, 10, 11], a strong effect of prior probability on response thresholds was observed in the current study. This was not only based on complexity penalized goodness of fit, which consistently favoured bias varying over the *B* parameter, but also in the posterior predictive analysis where omitting the threshold bias effect resulted in significant and consistent misfit. In both young and older adults, a significant biasing effect (i.e., incongruent threshold greater than congruent threshold) was observed in the block- and trial-wise biasing procedures, speaking to the robustness of the effect. Considering response caution (i.e., averaged incongruent and congruent parameter), older adults were more cautious than young adults, especially in the trial-wise biasing procedure. Given the well-established age-associated decline in cognitive processes such as processing speed and executive functioning [49], it is unsurprising that older adults were more cautious in the more labile and cognitively demanding trial-wise biasing condition.

However, the effect of prior probability was not entirely selective. That is, besides the canonical and well-reported effect on response thresholds, a significant effect was also observed on the nondecision-based process of response execution. In the current study, prior probability had a significant effect on response execution as evidenced by the best model always allowing bias to vary over the nondecision time parameter as well as the posterior predictive analysis exhibiting moderate misfit when omitting a bias effect on nondecision time. This robust effect was observed in both age groups and biasing conditions as the incongruent response execution time was greater (slower) than congruent response execution time. Recent research suggests an important role of GABAergic inhibitory pathways between the primary motor cortices in putatively subserving the observed response execution biasing effect [32]. Specifically, in the time period between the warning signal delivering prior probability information and the imperative signal, greater cortical inhibition was observed in the hand that was biased *away* from the cue (i.e., incongruent condition) than the hand that was biased towards the cue (i.e., congruent condition). Speculatively, this greater cortical inhibition is likely to have an influence by requiring more motoneurons and/or greater firing rates to generate a response, resulting in the observed increase in response execution time during the incongruent condition. Furthermore, the slower incongruent response execution time was most apparent in older adults in the trial-wise biasing condition. In addition to the aforementioned greater demands of the trial-wise bias condition and cognitive declines associated with healthy ageing, known age-related motor deficits such as movement slowing are likely to have played a role [27, 28].

Lastly, in the current study, we did not obtain convincing evidence for the influence of prior probability on the rate of evidence-accumulation. Though in the more challenging trial-wise biasing condition, and to a greater extent in older adults, some evidence of a rate effect was observed. The nature of this effect was small and inconsistent, most often exhibited in the opposite direction to that expected with a higher rate for the *less* likely outcome. In contrast, previous reports of prior probability effects on the quality of sensory evidence found that the more likely outcome has a higher rate of evidence-accumulation and the less likely outcome a lower rate [9, 12]. A potential explanation of this discrepancy is that these studies not only required more difficult perceptual discriminations than that required in our study but also utilised a different evidence-accumulation model (drift diffusion model vs. the racing diffusion model utilized here).

Future studies may build upon insights gained from the current research in numerous ways, some of which are outlined here. First, the generality of the observed non-selective effect of prior probability could be tested by investigating: i) other evidence-accumulation models such as the drift-diffusion model [2], ii) a range of prior probabilities (vs. only the 70% utilized

here), iii) other bias manipulations such as payoffs that often have similar canonical effects as prior probability [6], and iv) more difficult (error prone) choices. Second, computational approaches that jointly model the variation in behavioural *and* neurophysiological data could provide further insights into the neural mechanisms involved [50]. Lastly, given the differences observed here due to the healthy ageing process, the question of how these effects vary in clinical groups exhibiting cognitive and motor deficits is of importance.

In conclusion, the current study investigated, in groups of healthy young and older adults across different bias conditions, the selective influence assumption of prior probability using the racing diffusion model. Besides the expected influence of prior probability on response thresholds, a clear although smaller effect was also observed on the nondecision-based process of response execution time in both age groups and both bias conditions.

## Supporting information

**S1 Table. Median parameter estimates for the $Bt_0$ model.** Parameter values for both accumulators (LEFT: left response; RIGHT: right response) in the model where bias ('toleft' and 'toright' indicating leftward and rightward bias, respectively) was allowed to vary over unique combinations of threshold (B) and nondecision time ($t_0$ in seconds) for all age group (young and older adults) and bias type (block-wise and trial-wise bias) combinations, separately. In addition, the accumulator that matches and mismatches the stimulus is termed as 'v.true' and 'v.false', respectively.
(DOCX)

**S2 Table. Median parameter estimates for the $Bvt_0$ model.** Parameter values for both accumulators (LEFT: left response; RIGHT: right response) in the model where bias ('toleft' and 'toright' indicating leftward and rightward bias, respectively) was allowed to vary over unique combinations of threshold (B), accumulation rate (v), and nondecision time ($t_0$ in seconds) for all age group (young and older adults) and bias type (block-wise and trial-wise bias) combinations, separately. In addition, the accumulator that matches and mismatches the stimulus is termed as 'v.true' and 'v.false', respectively.
(DOCX)

**S1 Fig. Response time.** Observed (filled) and $Bvt_0$ model-predicted (unfilled) $10^{th}$ (square), $50^{th}$ (circle), and $90^{th}$ (triangle) error (top panel) and correct (bottom panel) response-time percentiles plotted, in milliseconds, separately for each age group–young (Y) and older (O) adults–as well as each bias type–block-wise (B) and trial-wise (T). Observed data points between congruent (C) and incongruent (IC) trial types are connected via dashed lines. Error bars for model-predicted response-times represent 95% credible intervals.
(DOCX)

**S2 Fig. Accuracy.** Observed (filled) and $Bvt_0$ model-predicted (unfilled) proportion of correct responses plotted separately for each age group–young (A) and older (B) adults–as well as each bias type–block-wise (1) and trial-wise (2). Observed data points between congruent (C) and incongruent (IC) trial types are connected via dashed lines. Error bars for model-predicted proportion of correct responses represent 95% credible intervals.
(DOCX)

**S3 Fig. Participant-level congruent response-time histograms.** For each participant, response times during the congruent (imperative signal congruent with warning signal) condition are plotted as a histogram (between 0 and 1000 ms with a 25 ms bin width, noting that

response times below 200 ms were discarded).
(DOCX)

## Acknowledgments

We would like to sincerely thank all the participants for contributing their valuable time.

## Author Contributions

**Conceptualization:** Rohan Puri, Mark R. Hinder, Andrew Heathcote.

**Data curation:** Rohan Puri, Andrew Heathcote.

**Formal analysis:** Rohan Puri, Andrew Heathcote.

**Funding acquisition:** Rohan Puri, Mark R. Hinder, Andrew Heathcote.

**Investigation:** Rohan Puri.

**Methodology:** Rohan Puri, Mark R. Hinder, Andrew Heathcote.

**Project administration:** Rohan Puri, Mark R. Hinder, Andrew Heathcote.

**Resources:** Mark R. Hinder, Andrew Heathcote.

**Software:** Rohan Puri, Andrew Heathcote.

**Supervision:** Mark R. Hinder, Andrew Heathcote.

**Validation:** Rohan Puri, Andrew Heathcote.

**Visualization:** Rohan Puri.

**Writing – original draft:** Rohan Puri.

**Writing – review & editing:** Rohan Puri, Mark R. Hinder, Andrew Heathcote.

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
