## [Decision Letter · Decision Letter 0]

30 Mar 2023

PONE-D-23-06495What mechanisms mediate prior probability effects on rapid-choice decision-making?PLOS ONE

Dear Dr. Puri,

Thank you for submitting your manuscript to PLOS ONE. After careful consideration, we feel that it has merit but does not fully meet PLOS ONE’s publication criteria as it currently stands. Therefore, we invite you to submit a revised version of the manuscript that addresses the points raised during the review process.

We look forward to receiving your revised manuscript.

Kind regards,

Thiago Fernandes, PhD

Academic Editor

PLOS ONE

Journal Requirements:

Reviewers' comments:

Reviewer's Responses to Questions

**Comments to the Author**

1. Is the manuscript technically sound, and do the data support the conclusions?

Reviewer #1: Yes

Reviewer #2: Yes

2. Has the statistical analysis been performed appropriately and rigorously? 

Reviewer #1: Yes

Reviewer #2: Yes

3. Have the authors made all data underlying the findings in their manuscript fully available?

Reviewer #1: Yes

Reviewer #2: Yes

4. Is the manuscript presented in an intelligible fashion and written in standard English?

Reviewer #1: Yes

Reviewer #2: No

5. Review Comments to the Author

Reviewer #1: The experimental paradigm employed in this study is similar to the Posner and/or Simon tasks, which are commonly used to explore the mechanisms of attention, but this study does not refer to attention but uses it as an intervention tool in the decision-making process. The paradigm that induces visual attention is used simply to manipulate decision bias. Based on these results, the model analysis reveals that prior probability affects more than just the reaction threshold portion of the accumulation of evidence in the decision-making process.

Overall, the study is based on a well-designed experimental design, and the analysis and discussion are for the most part valid. It clearly answers the question of whether prior probabilities affect nondecision processes, and this question is well supported by the data. This provides important evidence for the possibility that prior probabilities may also affect nondecision processes in the decision-making process, a possibility that has only been suggested by existing research. Therefore, this paper is worthy of inclusion in PLOS ONE.

Review comments are as follows;

* What do the numbers ('block-wise bias'; 2,3,5,7,8,10,11), ('trial-wise bias'; 4,9) mean?

* It appropriately addresses the problems that arise when analyzing reaction time data. In particular, the present study attempted to compare block-wise and trial-wise conditions, where the "fast guesses" are particularly likely to occur, and had to avoid the criticism that prior predictions in which the stimulus was not a factor were merely driven by the stimulus. In response, they 1) informed participants that the bias was not very high, 2) suppressed the incidence of fast guesses by mixing in catch trials, and 3) excluded the data of too short RTs that seemed to come from fast guesses with two types of filters: a threshold based on RTs themselves and an adaptive threshold based on correct response rate. The amount of data excluded by this operation is also reported, and the small number of excluded data (2.6%) indicates that the number of data is sufficient for the subsequent analysis. Furthermore, the fact that the fast guess is verified not only a priori, but also a posteriori, leaves no room to question whether the problem was really avoided.

* The effect of prior probabilities on the nondecision parameters of the decision-making process is examined by fitting reaction time data to the race diffusion model using a hierarchical Bayesian techniques. The initial values of each parameter, the constraints, the MCMC settings and convergence criteria, and the analysis software used are clearly described and detailed enough to replicate the analysis in this study given the data. However, it is not clear from the text alone how the bias effect parameter vbr was introduced as a term in the original race diffusion model, so it would be better if it were presented in an equation.

* The validity of the sub-model with a limited number of dependent parameters is tested using the devience evaluation criterion (DIC) to assess the influence of each parameter in each condition. According to [30], in the race diffusion model, the distribution of reaction times is determined by the threshold b, the evidence accumulation rate v, the nondecision time t0, and the distribution range A of initial values of sensory evidence, but in the present study, A is a constant independent of conditions, so they are interested in the bias effects of the first three. Although the procedure of ordering the relative importance of the parameters through model selection seems reasonable, we are not sure if the procedure of measuring the influence of the parameters by dividing the bias effect of the reduced model by the bias effect of the full model (Bvt0) is reasonable. As a result, overfitting occurs in the Bvt0 model, leading to the conclusion that the Bt0 model is reasonable. Considering this conclusion, using the overfitting Bvt0 as a standard does not seem at first glance to be a good approach.

* Although they judge the stimulus to be overfitting because of many negative bias effects, is it possible that negative bias effects are occurring due to the "inhibition of return (IOR)" of spatial attention (Posner & Cohen, 1984)? In the present stimulus, the directional symbols are slightly in the periphery, so there is an element of exogeneous cue. Therefore, it is possible that the stimulus is likely to induce IOR, in which participants are less likely to return their attention to the same position after a certain period of time has elapsed since the cue was presented. The total presentation delay from the start of priming to the start of target presentation is 1500 ms, and it is not surprising that this time interval would cause IOR. The Bvt0 model is not an overfitting but a result of correctly modeling IOR by parameter v, is it not? However, Figure 3 shows that, on average, no IOR is observed, so the conclusion that the Bt0 model looks more plausible appears to be correct. I would like to see a discussion on how this should be interpreted. The reason for the inconsistent results for parameter v may also be explained in terms of differences in the degree of IOR and individual differences due to differences in experimental conditions compared to existing studies.

* For Figures 3 and 4, the captions provide the necessary information, but the legends are not shown in the figures, which is unfriendly. In addition, the text should be a little larger for this resolution.

* Comparisons between conditions are presented in Table 5. The medians and 95% credible intervals of the bias effects clearly show the results of the tests. However, it takes some time to understand the meaning of the signs. It is understood from the caption of Table 5 that the bias effect is incongruent RT - congruent RT and the congruent RT is shorter when the effect is positive, but the result of the test of the difference of the bias effect does not clearly indicate which was subtracted from which, and we have to guess from the phrases in the text, like "older adults had slower but more accurate responses than young adults". In order to judge whether the inference in the text is correct or not, I would like to see an explanation in the caption so that the meaning of the sign can be understood.

* The discussion of the differences in average nondecision time (equivalent with response execution time in this study) between conditions, together with the neuroscientific evidence, is persuasive to a certain extent. However, I am not an expert in neuroscience, so I am not sure about the part "this greater cortical inhibition is likely to have an influence by requiring more motoneurons to generate a response." I do not know if the inference that the number of neurons (rather than the number of spikes) is necessary to counteract cortical inhibition is common in neuroscience. However, this explanation is an additional element and does not necessarily need to be reflected in the revision.

Reviewer #2: SEE THE ATTACHED DOCUMENT FOR MORE DETAILS

The study investigates the effect of prior probability on response thresholds using a racing diffusion model to account for response time and accuracy. They also investigated the effect of parameters on healthy younger versus healthy older adults. They found that the prior probability has a significant effect on response thresholds and clear influence on response execution across lifespan.

The introduction was clear with well explained reasoning. The Methods and Results explained in detail and Discussion In general, the sentences could be a bit shorter. That would make the reading easier. I have some minor comments:

Introduction:

• Page 5:

The sentence formation causes a confusion as to whether the prior probability on the nondecision time parameter is reflective only of reaction time and not response encoding. In the study, the manipulation is on the reaction time so it should be stated more clearly.

• Page7:

The false alarm rate for catch trials? The sentences are not very clear in this section. It would be good to clarify how response priming and perceptual priming is particularly tested and why.The paradigm is explained nicely in the methods part but it confuses the reader here.

Methods:

Line 347 prediciton -> prediction

Discussion:

• Starting Line 540

The first sentence should be reformulated. It is very hard to understand. More discussion is needed. Why was racing diffusion model chosen (compared to drift diffusion model for example) in explaining the effects of prior probability?

6. PLOS authors have the option to publish the peer review history of their article (what does this mean?). If published, this will include your full peer review and any attached files.

Reviewer #1: No

Reviewer #2: **Yes: **Isil Uluc

---

## [Author Response · Author response to Decision Letter 0]

18 Apr 2023

Reviewer #1

The experimental paradigm employed in this study is similar to the Posner and/or Simon tasks, which are commonly used to explore the mechanisms of attention, but this study does not refer to attention but uses it as an intervention tool in the decision-making process. The paradigm that induces visual attention is used simply to manipulate decision bias. Based on these results, the model analysis reveals that prior probability affects more than just the reaction threshold portion of the accumulation of evidence in the decision-making process.

Overall, the study is based on a well-designed experimental design, and the analysis and discussion are for the most part valid. It clearly answers the question of whether prior probabilities affect nondecision processes, and this question is well supported by the data. This provides important evidence for the possibility that prior probabilities may also affect nondecision processes in the decision-making process, a possibility that has only been suggested by existing research. Therefore, this paper is worthy of inclusion in PLOS ONE.

Review comments are as follows;

a) What do the numbers ('block-wise bias'; 2,3,5,7,8,10,11), ('trial-wise bias'; 4,9) mean?

• These numbers refer to the appropriate references, in line with PLOS ONE’s Vancouver style referencing guidelines, which utilised block-wise and trial-wise biasing procedures, respectively.

b) It appropriately addresses the problems that arise when analyzing reaction time data. In particular, the present study attempted to compare block-wise and trial-wise conditions, where the "fast guesses" are particularly likely to occur and had to avoid the criticism that prior predictions in which the stimulus was not a factor were merely driven by the stimulus. In response, they 1) informed participants that the bias was not very high, 2) suppressed the incidence of fast guesses by mixing in catch trials, and 3) excluded the data of too short RTs that seemed to come from fast guesses with two types of filters: a threshold based on RTs themselves and an adaptive threshold based on correct response rate. The amount of data excluded by this operation is also reported, and the small number of excluded data (2.6%) indicates that the number of data is sufficient for the subsequent analysis. Furthermore, the fact that the fast guess is verified not only a priori, but also a posteriori, leaves no room to question whether the problem was really avoided.

• We thank the reviewer for the positive feedback about our methodology to ensure fast guesses were excluded from analyses.

c) The effect of prior probabilities on the nondecision parameters of the decision-making process is examined by fitting reaction time data to the race diffusion model using a hierarchical Bayesian technique. The initial values of each parameter, the constraints, the MCMC settings and convergence criteria, and the analysis software used are clearly described and detailed enough to replicate the analysis in this study given the data. However, it is not clear from the text alone how the bias effect parameter vbr was introduced as a term in the original race diffusion model, so it would be better if it were presented in an equation.

• The equation has now been added to manuscript on Page 12, Line 277-284, and reads as,

“For example, if the rates for left and right accumulator are vleft and vright before the addition of bias, and denoting the minimum of these two values as dv (typically the rate for the mismatching accumulator), the rates after the addition of bias (with subscripts “right,bias” and “left,bias”) become:

vright,bias = vright + dv * vbr (a)

vleft,bias = vleft – dv * vbr (b)

Note that scaling vbr by dv ensures that the rates remain non-negative.”

Furthermore, in the spirit of open science, besides the description and equation in the manuscript, the source data and analyses code with model specifications have been uploaded to the Open Science Framework.

d) The validity of the sub-model with a limited number of dependent parameters is tested using the deviance evaluation criterion (DIC) to assess the influence of each parameter in each condition. According to [30], in the race diffusion model, the distribution of reaction times is determined by the threshold b, the evidence accumulation rate v, the nondecision time t0, and the distribution range A of initial values of sensory evidence, but in the present study, A is a constant independent of conditions, so they are interested in the bias effects of the first three. Although the procedure of ordering the relative importance of the parameters through model selection seems reasonable, we are not sure if the procedure of measuring the influence of the parameters by dividing the bias effect of the reduced model by the bias effect of the full model (Bvt0) is reasonable. As a result, overfitting occurs in the Bvt0 model, leading to the conclusion that the Bt0 model is reasonable. Considering this conclusion, using the overfitting Bvt0 as a standard does not seem at first glance to be a good approach.

• We would like to clarify that the influence of the parameters is considered by removing the bias effect on the parameter, and not the parameter itself. That is, the Bvt0 model where the bias effect of one parameter is omitted (by way of averaging or setting to 0), the Bvt0 model still includes all parameters. This ‘reduced model’ is then compared to the Bvt0 model where the bias effect of all parameters is included (‘full model’). Thus, there are no differences in parameters between the reduced and the full model and the Bt0 model is not the most reasonable purely in comparison to an overfitting Bvt0 model. This is stated on Page 15, Line 345-352, and reads as,

“Finally, for the overall best (i.e., the model with the lowest DIC value among all seven) and second-best models, the relative importance of each parameter type in simulating bias effects on two key behavioural measures of bias, response probability and correct RT, were evaluated by removing the bias effect on each parameter type while keeping it on the remaining parameter types [48]. For rate bias vbr was set to 0 for each posterior parameter sample. For B and t0 the four parameter estimates were reduced to two for each posteior sample by averaging over the bias manipulation.”

e) Although they judge the stimulus to be overfitting because of many negative bias effects, is it possible that negative bias effects are occurring due to the "inhibition of return (IOR)" of spatial attention (Posner & Cohen, 1984)? In the present stimulus, the directional symbols are slightly in the periphery, so there is an element of exogeneous cue. Therefore, it is possible that the stimulus is likely to induce IOR, in which participants are less likely to return their attention to the same position after a certain period of time has elapsed since the cue was presented. The total presentation delay from the start of priming to the start of target presentation is 1500 ms, and it is not surprising that this time interval would cause IOR. The Bvt0 model is not an overfitting but a result of correctly modeling IOR by parameter v, is it not? However, Figure 3 shows that, on average, no IOR is observed, so the conclusion that the Bt0 model looks more plausible appears to be correct. I would like to see a discussion on how this should be interpreted. The reason for the inconsistent results for parameter v may also be explained in terms of differences in the degree of IOR and individual differences due to differences in experimental conditions compared to existing studies.

• The reviewer makes a good suggestion regarding Inhibition of Return (IOR) underlying the negative evidence accumulation parameters. However, the observed data in Figure 3, as highlighted by the reviewer, does not provide support for this. Specifically, reaction time is not slower for congruent trials (where inhibition of return would be most prevalent) with this pattern observed in both bias types (block-wise and trial-wise) as well as both age groups (young and older adults). Thus, from our perspective, overfitting is the most likely explanation of the negative bias effects in the Bvt0 model (as stated on Page 19, Line 392).

f) For Figures 3 and 4, the captions provide the necessary information, but the legends are not shown in the figures, which is unfriendly. In addition, the text should be a little larger for this resolution.

• To ensure the relevant data was highlighted to the highest degree, the choice was made to not overcrowd the figures with additional text via in-figure legends and to instead have all the relevant information in the figure captions.

g) Comparisons between conditions are presented in Table 5. The medians and 95% credible intervals of the bias effects clearly show the results of the tests. However, it takes some time to understand the meaning of the signs. It is understood from the caption of Table 5 that the bias effect is incongruent RT - congruent RT and the congruent RT is shorter when the effect is positive, but the result of the test of the difference of the bias effect does not clearly indicate which was subtracted from which, and we have to guess from the phrases in the text, like "older adults had slower but more accurate responses than young adults". In order to judge whether the inference in the text is correct or not, I would like to see an explanation in the caption so that the meaning of the sign can be understood.

• The caption of Table 5 has now been amended in line with the reviewer’s feedback with the addition of the below passage on Page 27 (Line 498-501),

“For block- and trial-wise bias types, positive values indicate a larger incongruent, than congruent, value, whereas for block-wise vs. trial-wise bias type and young vs. older adult comparisons, positive values indicate a greater bias effect in the former.”

h) The discussion of the differences in average nondecision time (equivalent with response execution time in this study) between conditions, together with the neuroscientific evidence, is persuasive to a certain extent. However, I am not an expert in neuroscience, so I am not sure about the part "this greater cortical inhibition is likely to have an influence by requiring more motoneurons to generate a response." I do not know if the inference that the number of neurons (rather than the number of spikes) is necessary to counteract cortical inhibition is common in neuroscience. However, this explanation is an additional element and does not necessarily need to be reflected in the revision.

• The reviewer makes a valid point about the number of spikes, and this has now been added to our speculation (Page 29, Line 543), which reads as,

“… by requiring more motoneurons and/or greater firing rates…”

Reviewer #2

The study investigates the effect of prior probability on response thresholds using a racing diffusion model to account for response time and accuracy. They also investigated the effect of parameters on healthy younger versus healthy older adults. They found that the prior probability has a significant effect on response thresholds and clear influence on response execution across lifespan.

The introduction was clear with well explained reasoning. The Methods and Results explained in detail and Discussion In general, the sentences could be a bit shorter. That would make the reading easier. I have some minor comments:

a) Page 5: Firstly, it was reasoned that (any) effects of prior probability on the nondecision time parameter was reflective of effects on the response execution time, and not the stimulus encoding time, component. Given the identical optical properties of the stimuli in the current study (see ‘Experimental procedure’ section) and that stimulus encoding extracts decision-relevant evidence, it is highly unlikely that the stimulus encoding time component is different for the left vs. right accumulators.

The sentence formation causes a confusion as to whether the prior probability on the nondecision time parameter is reflective only of reaction time and not response encoding. In the study, the manipulation is on the reaction time so it should be stated more clearly.

• The first sentence, on Page 5 (Line 107-109), states that any prior probability effects on the nondecision time parameter are assumed to be reflective on the response execution component and not the stimulus encoding time component. The next sentence highlights why stimulus encoding would be expected to not be affected with the line following that reasoning as to why response execution would be expected to be affected.

b) Page7: (‘prime’), regardless of its validity, prior to the imperative 137 signal (‘target’) might invoke response priming with the visual similarity between the warning and imperative signal possibly invoking perceptual priming (see Fig 2 and ‘Experimental procedure’ section). Response priming in the current study was tested with the presence of ‘catch’ trials (i.e., trials where the warning signal was not followed by an imperative signal) such that a negligible influence of response priming would be reflected by participants correctly withholding their responses on catch trials. Perceptual priming was tested by ensuring that the warning signal prime and the imperative signal target utilized the same symbol (arrows as per Fig 2), with any perceptual priming effects biasing sensory evidence and being reflected as superior model fits of consistent evidence-accumulation parameters (v parameter in Fig 1).

The false alarm rate for catch trials? The sentences are not very clear in this section. It would be good to clarify how response priming and perceptual priming is particularly tested and why. The paradigm is explained nicely in the methods part, but it confuses the reader here.

• Indeed, this is the false alarm rate for catch trials and has now been added to the manuscript in areas where this is being defined. In regard to why response and perceptual priming are tested, this is stated on Page 6, Line 134-136, and reads as follows,

“However, one may argue that the different priming of left vs. right responses, and more broadly decision-based processes as well, may be influenced by other mechanisms besides prior probability”.

c) Line 347 prediciton -> prediction

• We thank the reviewer for picking up on this typographical error, which has now been amended.

d) Line 540: In contrast, previous reports that prior probability affects the quality of sensory evidence found that the more likely outcome has a higher rate of evidence-accumulation and the less likely outcome a lower rate [9,12]. A potential explanation of this discrepancy is that these studies not only required more difficult perceptual discriminations than that required in our study but also utilised a different evidence accumulation model (drift diffusion model vs. the racing diffusion model utilized here).

The first sentence should be reformulated. It is very hard to understand. More discussion is needed. Why was racing diffusion model chosen (compared to drift diffusion model for example) in explaining the effects of prior probability?

• This has now been rewritten for clarity and reads as,

“In contrast, previous reports of prior probability effects on the quality of sensory evidence found that the more likely outcome has a higher rate of evidence-accumulation and the less likely outcome a lower rate [9,12]”

---

## [Decision Letter · Decision Letter 1]

5 May 2023

PONE-D-23-06495R1What mechanisms mediate prior probability effects on rapid-choice decision-making?PLOS ONE

Dear Dr. Puri,

Thank you for submitting your manuscript to PLOS ONE. After careful consideration, we feel that it has merit but does not fully meet PLOS ONE’s publication criteria as it currently stands. Therefore, we invite you to submit a revised version of the manuscript that addresses the points raised during the review process.

We look forward to receiving your revised manuscript.

Kind regards,

Thiago P. Fernandes, PhD

Academic Editor

PLOS ONE

Journal Requirements:

Additional Editor Comments:

Thank you for your valuable submission, thoughtful and careful edits.

Please, along with reviewer’s comments, address a few things

1. Double check grammar, there are extensive sections that can be shortened and others that paragraph has two ideas interchangeable at same time;

2. Check your references accordingly to the Journal’s requirements - keep them constant to help typesetting;

3. Remember to detail all stats parameters and reason to pick up them

Wishing you success with this very good study.

Reviewers' comments:

Reviewer's Responses to Questions

**Comments to the Author**

1. If the authors have adequately addressed your comments raised in a previous round of review and you feel that this manuscript is now acceptable for publication, you may indicate that here to bypass the “Comments to the Author” section, enter your conflict of interest statement in the “Confidential to Editor” section, and submit your "Accept" recommendation.

Reviewer #1: (No Response)

Reviewer #2: All comments have been addressed

2. Is the manuscript technically sound, and do the data support the conclusions?

Reviewer #1: Yes

Reviewer #2: Yes

3. Has the statistical analysis been performed appropriately and rigorously? 

Reviewer #1: Yes

Reviewer #2: Yes

4. Have the authors made all data underlying the findings in their manuscript fully available?

Reviewer #1: Yes

Reviewer #2: Yes

5. Is the manuscript presented in an intelligible fashion and written in standard English?

Reviewer #1: Yes

Reviewer #2: Yes

6. Review Comments to the Author

Reviewer #1: >These numbers refer to the appropriate references, in line with PLOS ONE’s Vancouver style referencing guidelines, which utilised block-wise and trialwise biasing procedures, respectively.

After opening manuscript.docx, I finally understood that "('block-wise bias'; 2,3,5,7,8,10,11)" and "('trial-wise bias'; 4,9)" in lines 82-83 are the citation numbers. The peer-review PDF I read had no links to the references, no bracket [], and no explanation of the numbers, so I did not know what the numbers were at first. The Vancouver style recommends brackets [] or superscripts to indicate that the number is a bibliography number. In the published version of the PDF, the numbers are linked to the bibliography, so the effect is minor, but we recommend bracketing the numbers for better clarity. For example, "('block-wise bias'; [2,3,5,7,8,10,11])".

>We thank the reviewer for the positive feedback about our methodology to ensure fast guesses were excluded from analyses.

I thought this methodology was a good example of analyzing reaction times and one of the most commendable aspects of the paper.

>The equation has now been added to manuscript on Page 12, Line 277-284, and reads as, “For example, if the rates for left and right accumulator are vleft and vright before the addition of bias, and denoting the minimum of these two values as dv (typically the rate for the mismatching accumulator), the rates after the addition of bias (with subscripts “right,bias” and “left,bias”) become:

>vright,bias = vright + dv * vbr (a)

>vleft,bias = vleft – dv * vbr (b)

>Note that scaling vbr by dv ensures that the rates remain non-negative.”

>Furthermore, in the spirit of open science, besides the description and equation in the manuscript, the source data and analyses code with model specifications have been uploaded to the Open Science Framework.

Regarding the Model fitting section of the Racing Diffusion Model, this modification makes it easier to understand how the bias effect parameter vbr is specifically implemented, since it is now shown in a mathematical expression. I also confirmed that the analysis data and the source code of R can be downloaded; I wonder if the bias effect in parameters other than v (i.e., B, t0) can also be expressed by the same equation? Or is it possible that only the parameter v is treated specially, since the symbols for the bias parameters for B and t0 are not mentioned in any of the following explanations. A word of explanation on this point would be appreciated.

However, the bias effect parameter explanation should be written more clearly as it confuses the reader. From lines 325-328, I can read that two bias effect parameters are added, and each parameter express that the warning signal is left or right. Since the Btv0 model has 12 parameters, I can enumerate all the parameters of the Bvt0 model, that is, (B_l, B_r, v_l, v_r, t0_l, t0_r, Bbr_l, Bbr_r, vbr_l, vbr_r, t0br_l, t0br_r), where the variables with suffix "br_x" are eliminated in the degenerate model. Here, "_l, _r" are suffices indicating the direction of response or the left and right sides of the accumulator, and "br_l, br_r" are suffices indicating the bias effect when the warning signal is to the left and to the right, respectively. On the other hand, I understood that the formula added in lines 277-284 has only one bias effect parameter, vbr (dv is not a free parameter since it selects the smaller of vright and vleft, and that the common variable vbr is used for v(left,bias) and v(right,bias)). The latter explanation seems to express the left and right bias directions by the sign of vbr and the Sampling section (lines 297-300) also describes only vbr as a bias effect parameter. Therefore, it is seemingly contradictory.

>We would like to clarify that the influence of the parameters is considered by removing the bias effect on the parameter, and not the parameter itself. That is, the Bvt0 model where the bias effect of one parameter is omitted (by way of averaging or setting to 0), the Bvt0 model still includes all parameters. This ‘reduced model’ is then compared to the Bvt0 model where the bias effect of all parameters is included (‘full model’). Thus, there are no differences in parameters between the reduced and the full model and the Bt0 model is not the most reasonable purely in comparison to an overfitting Bvt0 model. This is stated on Page 15, Line 345-352, and reads as, ...(omitted below)

To calculate the ratio of bias effects, the author uses a "reduced model" (e.g., the Bt0 model) in which certain parameters are modeled as mean values (or 0) independent of the bias direction (not being removed), and a "full model" (Btv0 model) in which all three parameters are modeled as variables dependent on the bias direction, where all three parameters are modeled as variables that depend on the bias direction. The ratio of the bias effect is the bias effect of the reduced model divided by the bias effect of the full model. Since both models are complete in terms of the parameter types themselves, indeed, this ratio is an indicator of the relative importance of the parameters in the Btv0 model. It is still questionable whether the overfitting Btv0 model can be used as the basis for the indicator, but it is only one of the indicators in this paper. This issue is not critical since the decision is made in combination with other indicators such as the DIC, which calculates the complexity of the model as a cost.

>The reviewer makes a good suggestion regarding Inhibition of Return (IOR) underlying the negative evidence accumulation parameters. However, the observed data in Figure 3, as highlighted by the reviewer, does not provide support for this. Specifically, reaction time is not slower for congruent trials (where inhibition of return would be most prevalent) with this pattern bserved in both bias types (block-wise and trial-wise) as well as both age groups (young and older adults). Thus, from our perspective, overfitting is the most likely explanation of the negative bias effects in the Bvt0 model (as stated on Page 19, Line 392).

The author's argument is valid. If there is an effect of inhibition of return (IOR), the reaction time should clearly slow down even in the matching condition. However, Figure 3, which is close to the raw data, shows that the reaction time actually consistently speeded up in the matching direction. This makes it difficult to believe that IOR was occurring and that it was correctly modeled by the Bvt0 model.

>To ensure the relevant data was highlighted to the highest degree, the choice was made to not overcrowd the figures with additional text via in-figure legends and to instead have all the relevant information in the figure captions.

Yes, I understand. I prefer to have the plot legend in the figure image, but the PLOS ONE guidelines allow authors to decide whether to include the legend in the figure or not. The text is clearly visible by referring to the original tif file, so there is no problem here.

>The caption of Table 5 has now been amended in line with the reviewer’s feedback with the addition of the below passage on Page 27 (Line 498-501), “For block- and trial-wise bias types, positive values indicate a larger incongruent, than congruent, value, whereas for block-wise vs. trial-wise bias type and young vs. older adult comparisons, positive values indicate a greater bias effect in the former.”

Thanks for the additional passage that made it easier to understand.

>The reviewer makes a valid point about the number of spikes, and this has now been added to our speculation (Page 29, Line 543), which reads as, “… by requiring more motoneurons and/or greater firing rates…”

Thank you very much for reflecting my point.

Reviewer #2: The authors responded all comments satisfactorily. The language was clear and the data analysis and results were explained clearly.

I thank the authors for their hard work.

I have no further comments or suggestions.

7. PLOS authors have the option to publish the peer review history of their article (what does this mean?). If published, this will include your full peer review and any attached files.

Reviewer #1: No

Reviewer #2: **Yes: **Isil Uluc

---

## [Author Response · Author response to Decision Letter 1]

13 Jun 2023

Editor

Thank you for your valuable submission, thoughtful and careful edits.

Please, along with reviewer’s comments, address a few things

a) Double check grammar, there are extensive sections that can be shortened and others that paragraph has two ideas interchangeable at same time.

The opportunity was taken to double check grammar, shorten sentences, and ensure paragraphs have a consistent flow of ideas.

b) Check your references accordingly to the Journal’s requirements - keep them constant to help typesetting.

References have been checked, ensuring they are as per the journal’s requirements.

c) Remember to detail all stats parameters and reason to pick up them

This has been detailed and reasoned to the best of our ability.

Wishing you success with this very good study.

Reviewer #1

a) These numbers refer to the appropriate references, in line with PLOS ONE’s Vancouver style referencing guidelines, which utilised block-wise and trialwise biasing procedures, respectively.

After opening manuscript.docx, I finally understood that "('block-wise bias'; 2,3,5,7,8,10,11)" and "('trial-wise bias'; 4,9)" in lines 82-83 are the citation numbers. The peer-review PDF I read had no links to the references, no bracket [], and no explanation of the numbers, so I did not know what the numbers were at first. The Vancouver style recommends brackets [] or superscripts to indicate that the number is a bibliography number. In the published version of the PDF, the numbers are linked to the bibliography, so the effect is minor, but we recommend bracketing the numbers for better clarity. For example, "('block-wise bias'; [2,3,5,7,8,10,11])".

The references now have square brackets for additional clarity.

b) We thank the reviewer for the positive feedback about our methodology to ensure fast guesses were excluded from analyses.

I thought this methodology was a good example of analyzing reaction times and one of the most commendable aspects of the paper.

We thank the reviewer for the positive feedback.

c) The equation has now been added to manuscript on Page 12, Line 277-284, and reads as, “For example, if the rates for left and right accumulator are vleft and vright before the addition of bias, and denoting the minimum of these two values as dv (typically the rate for the mismatching accumulator), the rates after the addition of bias (with subscripts “right,bias” and “left,bias”) become:

>vright,bias = vright + dv * vbr (a)

>vleft,bias = vleft – dv * vbr (b)

>Note that scaling vbr by dv ensures that the rates remain non-negative.”

>Furthermore, in the spirit of open science, besides the description and equation in the manuscript, the source data and analyses code with model specifications have been uploaded to the Open Science Framework.

Regarding the Model fitting section of the Racing Diffusion Model, this modification makes it easier to understand how the bias effect parameter vbr is specifically implemented, since it is now shown in a mathematical expression. I also confirmed that the analysis data and the source code of R can be downloaded; I wonder if the bias effect in parameters other than v (i.e., B, t0) can also be expressed by the same equation? Or is it possible that only the parameter v is treated specially, since the symbols for the bias parameters for B and t0 are not mentioned in any of the following explanations. A word of explanation on this point would be appreciated.

However, the bias effect parameter explanation should be written more clearly as it confuses the reader. From lines 325-328, I can read that two bias effect parameters are added, and each parameter express that the warning signal is left or right. Since the Btv0 model has 12 parameters, I can enumerate all the parameters of the Bvt0 model, that is, (B_l, B_r, v_l, v_r, t0_l, t0_r, Bbr_l, Bbr_r, vbr_l, vbr_r, t0br_l, t0br_r), where the variables with suffix "br_x" are eliminated in the degenerate model. Here, "_l, _r" are suffices indicating the direction of response or the left and right sides of the accumulator, and "br_l, br_r" are suffices indicating the bias effect when the warning signal is to the left and to the right, respectively. On the other hand, I understood that the formula added in lines 277-284 has only one bias effect parameter, vbr (dv is not a free parameter since it selects the smaller of vright and vleft, and that the common variable vbr is used for v(left,bias) and v(right,bias)). The latter explanation seems to express the left and right bias directions by the sign of vbr and the Sampling section (lines 297-300) also describes only vbr as a bias effect parameter. Therefore, it is seemingly contradictory.

It is only the v parameter that is treated according to the mathematical expression. For the B and t0 parameters, bias effects for each accumulator are as follows using the reviewer’s notation,

Bbr_l, Bbr_r, Bbl_l, Bbl_r, t0br_l, t0br_r, t0bl_l, and t0_bl_r (8 parameters)

This is clarified in the manuscript and reads as,

“Threshold and nondecision time were parameterised as b and t0, respectively, for every bias direction (left and right) and accumulator (left and right) combination.”

For v, the 4 parameters are v.true (matching accumulator), v.false (mismatching accumulator), vbr_l (vBR parameter for the leftward pointing warning signal), and vbr_r (vBR parameter for the rightward pointing warning signal). Thus, as the reviewer pointed out, vbr_l would be expected to be negative and vbr_r would be expected to be positive.

All parameters for all the models are stated in the ‘model_create.R’ file.

d) We would like to clarify that the influence of the parameters is considered by removing the bias effect on the parameter, and not the parameter itself. That is, the Bvt0 model where the bias effect of one parameter is omitted (by way of averaging or setting to 0), the Bvt0 model still includes all parameters. This ‘reduced model’ is then compared to the Bvt0 model where the bias effect of all parameters is included (‘full model’). Thus, there are no differences in parameters between the reduced and the full model and the Bt0 model is not the most reasonable purely in comparison to an overfitting Bvt0 model. This is stated on Page 15, Line 345-352, and reads as, ...(omitted below)

To calculate the ratio of bias effects, the author uses a "reduced model" (e.g., the Bt0 model) in which certain parameters are modeled as mean values (or 0) independent of the bias direction (not being removed), and a "full model" (Btv0 model) in which all three parameters are modeled as variables dependent on the bias direction, where all three parameters are modeled as variables that depend on the bias direction. The ratio of the bias effect is the bias effect of the reduced model divided by the bias effect of the full model. Since both models are complete in terms of the parameter types themselves, indeed, this ratio is an indicator of the relative importance of the parameters in the Btv0 model. It is still questionable whether the overfitting Btv0 model can be used as the basis for the indicator, but it is only one of the indicators in this paper. This issue is not critical since the decision is made in combination with other indicators such as the DIC, which calculates the complexity of the model as a cost.

Besides the use of other indicators, such as the DIC, we must clarify that the ‘reduced’ and ‘full’ models are not the Bt0 model with certain parameters modelled as mean values or 0 and the Bvt0 model in which all three parameters are modelled to be dependent on bias direction, respectively. The ‘full’ models are the Bvt0 model and the Bt0 model where the parameters vary by bias direction whereas the ‘reduced’ models are where the parameters are modelled as mean values or 0. This is presented separately for the Bvt0 model (Table 2) and the Bt0 model (Table 3) in the Results section of the manuscript.

c) The reviewer makes a good suggestion regarding Inhibition of Return (IOR) underlying the negative evidence accumulation parameters. However, the observed data in Figure 3, as highlighted by the reviewer, does not provide support for this. Specifically, reaction time is not slower for congruent trials (where inhibition of return would be most prevalent) with this pattern bserved in both bias types (block-wise and trial-wise) as well as both age groups (young and older adults). Thus, from our perspective, overfitting is the most likely explanation of the negative bias effects in the Bvt0 model (as stated on Page 19, Line 392).

The author's argument is valid. If there is an effect of inhibition of return (IOR), the reaction time should clearly slow down even in the matching condition. However, Figure 3, which is close to the raw data, shows that the reaction time actually consistently speeded up in the matching direction. This makes it difficult to believe that IOR was occurring and that it was correctly modeled by the Bvt0 model.

We thank the reviewer for their confirmation of our response.

d) To ensure the relevant data was highlighted to the highest degree, the choice was made to not overcrowd the figures with additional text via in-figure legends and to instead have all the relevant information in the figure captions.

Yes, I understand. I prefer to have the plot legend in the figure image, but the PLOS ONE guidelines allow authors to decide whether to include the legend in the figure or not. The text is clearly visible by referring to the original tif file, so there is no problem here.

We thank the reviewer for their confirmation of our response.

e) The caption of Table 5 has now been amended in line with the reviewer’s feedback with the addition of the below passage on Page 27 (Line 498-501), “For block- and trial-wise bias types, positive values indicate a larger incongruent, than congruent, value, whereas for block-wise vs. trial-wise bias type and young vs. older adult comparisons, positive values indicate a greater bias effect in the former.”

Thanks for the additional passage that made it easier to understand.

We thank the reviewer for their appreciation of our response.

f) The reviewer makes a valid point about the number of spikes, and this has now been added to our speculation (Page 29, Line 543), which reads as, “… by requiring more motoneurons and/or greater firing rates…”

Thank you very much for reflecting my point.

We thank the reviewer for their appreciation of our response.

Reviewer #2

The authors responded all comments satisfactorily. The language was clear, and the data analysis and results were explained clearly.

I thank the authors for their hard work.

I have no further comments or suggestions.

 We are pleased that our responses were to the reviewer’s satisfaction.

---

## [Editor Report · Decision Letter 2]

19 Jun 2023

What mechanisms mediate prior probability effects on rapid-choice decision-making?

PONE-D-23-06495R2

Dear Dr. Puri,

We’re pleased to inform you that your manuscript has been judged scientifically suitable for publication and will be formally accepted for publication once it meets all outstanding technical requirements.

Kind regards,

Thiago P. Fernandes, PhD

Academic Editor

PLOS ONE
---

## [Editor Report · Acceptance letter]

29 Jun 2023

PONE-D-23-06495R2 

What mechanisms mediate prior probability effects on rapid-choice decision-making? 

Dear Dr. Puri:

I'm pleased to inform you that your manuscript has been deemed suitable for publication in PLOS ONE. Congratulations! Your manuscript is now with our production department. 

Kind regards, 

on behalf of

Dr. Thiago P. Fernandes 

Academic Editor

PLOS ONE